# A Survey of Collaborative UAV–WSN Systems for Efficient Monitoring

**DOI:** 10.3390/s19214690

**Published:** 2019-10-28

**Authors:** Dan Popescu, Florin Stoican, Grigore Stamatescu, Oana Chenaru, Loretta Ichim

**Affiliations:** 1Faculty of Automatic Control and Computers, University POLITEHNICA of Bucharest, 060042 Bucharest, Romania; florin.stoican@upb.ro (F.S.); grigore.stamatescu@upb.ro (G.S.); oana.chenaru@upb.ro (O.C.); iloretta@yahoo.com (L.I.); 2Institute of Technical Informatics, Graz University of Technology, 16 Inffeldgasse, 8010 Graz, Austria

**Keywords:** unmanned aerial system, wireless sensor network, data acquisition, intelligent data processing, communication protocol and standardization, data transmission, internet of things, integrated collaborative systems, path design

## Abstract

Integrated systems based on wireless sensor networks (WSNs) and unmanned aerial vehicles (UAVs) with electric propulsion are emerging as state-of-the-art solutions for large scale monitoring. Main advances stemming both from complex system architectures as well as powerful embedded computing and communication platforms, advanced sensing and networking protocols have been leveraged to prove the viability of this concept. The design of suitable algorithms for data processing, communication and control across previously disparate domains has thus currently become an intensive area of interdisciplinary research. The paper was focused on the collaborative aspects of UAV–WSN systems and the reference papers were analyzed from this point of view, on each functional module. The paper offers a timely review of recent advances in this area of critical interest with focus on a comparative perspective across multiple recent theoretical and applied contributions. A systematic approach is carried out in order to structure a unitary from conceptual design towards key implementation aspects. Focus areas are identified and discussed such as distributed data processing algorithms, hierarchical multi-protocol networking aspects and high level WSN–constrained UAV-control. Application references are highlighted in various domains such as environmental, agriculture, emergency situations and homeland security. Finally, a research agenda is outlined to advance the field towards tangible economic and social impact.

## 1. Introduction

The tendency to use collaborative unmanned aerial vehicle-wireless sensor network (UAV–WSN) systems for surveillance, exploring and monitoring large regions of interest is increasingly strong today. These systems are able to integrate information from ground (WSN), air (UAV), space (satellite—GPS or equivalent) and internet. The advantage of these collaborative systems has been emphasized for large-scale monitoring [1], increasing mobility, accessibility and reaction time in case of emergency. The main novelty included a hybrid system architecture for data collection, which integrates an optimal method for the UAV trajectory design in conjunction with cluster head selection schemes at the ground sensor level. Practical development and implementation allow validation of the proposed theoretical approach in scalable fashion with tens to hundreds of ground nodes. Field level data is first aggregated at the node level before being relayed through the UAV systems to the central gateway. Salient example of data aggregation methods is represented by consensus algorithms, which allow for distributed agreement locally thereby reducing the burden on the communication channels and reducing overall latency. Matching WSN transmission schedules with UAV over flight patterns is also highly relevant for the collaborative aspects of such large-scale monitoring systems. Neither of the subsystems is efficient in what regards data transmission, reliability and duration of functioning by itself. Managing communication only through ground sensors would quickly deplete their batteries and using only UAVs would be wasteful from an energy point of view. Thus, a mix between these two types of agents (fixed and mobile) provides a solution, which is better than the sum of their parts.

Alongside this initial example there are many relevant and recent contributions, mostly in the form of papers, research projects and commercial products concerning either WSN or UAV implementations. Neither of these subsystems is efficient in what regards data transmission, reliability and duration of functioning by itself. Managing communication only through ground sensors would quickly deplete their batteries and using only UAVs would be wasteful from an energy point of view. Thus, a mix between these two types of agents (fixed and mobile) provides a solution, which is better than the sum of their parts.

Several surveys are also available that synthesize the characteristics, application areas and challenges in each of them, such as [2,3,4,5,6] for WSN and [7,8] for UAV. Over time, sensor networks have been widely used for data collection in both outdoor and indoor settings and in various application domains. Gradually, given significant technological development, the sensor networks have been enhanced with data processing elements, towards on-board intelligence. WSN design considerations, mainly relating to: production cost, accessibility, scalability and reliability, networking protocols, physical constraints such as: energy consumption, communication range and autonomy, data transmission and processing capabilities have been analyzed in many papers in the last years. The reference WSN architecture has a hierarchical tree structure and includes several sensing nodes (SN) grouped around a cluster head (CH). The CH is responsible for collecting and aggregating sensor data to make it available to the user [1]. More complex architectures focus on optimizing packet transmission time and reliability and involve multi-hop, mesh routing and even adaptive and dynamic topologies, such as in cellular networks, where challenge arise as nodes can join or leave the network at any time.

On the other hand, unmanned aerial vehicles experienced a rapid development as they enabled fast access to otherwise inaccessible areas, with initial focus in military and disaster management fields. They were referred in the literature under various names such as: UAVs, drones, aerial platforms and aerial robots. UAVs can be characterized according to different categories: (a) according to the type of communication: open, specific and certified—EASA (European Aviation Safety Agency) classification [9]; (b) according to the type of motor: with thermal or electric motors and (c) according to the type of wing: airplane type (fixed wing) or helicopter type (rotor or multi-copter). The EASA rules, which are based on the evaluation of the risk of operation, establishes the obligations of drone manufacturers and operators in terms of safety, respect for privacy, the environment, protection against noise and security. Thus, starting from June 2020 all drone operators shall register themselves and receive an authorization before using a UAV. The registration number needs to be displayed on the UAV. It must be specified if the UAV path is below 120 m (visual line of sight—VLOS) or more (beyond visual line of sight—BVLOS). Increased pay-load capabilities have allowed equipping them with cameras to easily survey a given area, with small tanks of chemical solutions for spraying pesticides or fire extinguishing and even with WSN nodes for faster deployment. While the basic architecture considers a single UAV and one or more base stations for transmitting trajectory parameters, more recent advances analyze multi-copter applications, with or without inter-UAV cooperation to overcome energy and distance limitations. Probably the most important issue is that these devices raised safety concerns regarding the allocation of airspace, which is currently separated from that of commercial airplanes or helicopters. This problem is under analysis at both research and institutional/regulatory levels and it is expected that by using specific intelligent technologies to reduce the risk both in the air and on ground, a suitable and widely adopted solution will be identified [9].

While neither WSN nor UAV architectures lack individual applicability, in the last years researchers showed interest in integrating these technologies. Motivated by the current evolving requirements for monitoring and data collection from extensive geographical spaces, collaborative, hybrid (air-ground or mobile-fixed) UAV–WSN systems have been developed [1,10]. The efficient use in wide areas of the UAV–WSN collaborative systems, however, requires an artificial intelligence component for the acquisition, transmission and processing of data as well as for the control of the mission. Although commercially UAVs with different propulsion systems are available, in this paper we will only have under consideration those with electric propulsion, which are generally small, with reduced maintenance and operational costs. This is also aligned with the current regulatory context in Europe and beyond concerning the reduction of carbon emissions, which is driving electrification in the transport sector. Electric cars replacing gas-powered vehicles represent just one salient example. For most monitoring applications, the use of UAV with electric propulsion is currently widespread. Considering the reduced flight autonomy in comparison to thermal powered versions, one solution to extend the useful coverage area is to deploy multi-UAV systems. On the other hand, satellites are currently indispensable for most global navigation and other applications, but they can also be leveraged as complementary support infrastructure for transmitting data or commands for the analyzed UAV–WSN systems. Their impact on the environment however should also be accounted for, especially due to the release of small particles by rockets at launched [11].

This paper aims to review and highlight the collaborative aspect between the mentioned elements, organized in an integrated UAV–WSN architecture, as well as the intelligent data processing, computation, communication and control methods for carrying out specific monitoring and evaluation tasks in certain fields such as: transport, agriculture, disaster management, environmental assessment, etc.

The paper topic is broad, including many keywords, but they must be viewed from the standpoint of the collaborative, integrated UAV–WSN system, as they are a means of integration and collaboration between ground agents (WSN) and air agents (UAV). On the other hand, the ways in which the different functional components of the system collaborate with each other were highlighted. For example, the motion planning procedures are discussed from the viewpoint of their interaction with the rest of the WSN: path with sensory data collection, image acquisition and processing, efficient communication with data collection and processing, etc.

Recently, many studies have been carried out on the integration of UAV–WSN systems into the Internet of things (IoT) paradigm for long-distance mission communication and control or for cloud processing and storage. With the increasing number of papers in IoT technologies and considering the IoT role as a data sensing and/or actuating node, we believe it is of great interest including this topic in our research. Not the least important, a special emphasis was placed on distributed computing elements stemming from Fog and Edge Computing [12,13,14] domains with direct relevance to the efficient design of collaborative UAV–WSN systems.

Many recent contributions, in the form of papers, research projects, and commercial products concern the implementation of WSN [2] and UAV [3] systems. The contribution and timely aspect of our article stem from a focused overview and highlight regarding the key collaborative aspects between the mentioned elements as well as the intelligent data processing, computation, communication and control methods. These lead to carrying out specific monitoring and evaluation tasks in application domains of critical interest such as: transport, agriculture, natural disaster damage management, environmental assessment and others.

Over time, sensor networks have been widely used for data collection both outdoor and indoor, in various fields of applications [3,4,5]. Gradually, due to the technological development, the sensor networks were equipped with data processing elements, thus becoming intelligent ones [6].

In our vision, the blueprint for such a complex, collaborative, monitoring system is composed of four main parts, as illustrated by Figure 1: Ground WSNs, UAVs, Internet/global system for mobile communications (GSM) networks and ground control stations (GCSs). From the communication point of view, we identified five possible communication channels: WSN–WSN (between SN—marked with red and CH marked with blue), WSN–UAV (between CH and WSN), UAV-GCS (between UAV and GDT, associated with GCS), GCS-internet-GCS, UAV–UAV and UAV-satellite-GCS. The primary task of the UAV team is to cover a given area where one or several WSNs were deployed, having as main objectives collecting data and/or capturing surface images from a certain area. The main constraints for accomplishing these tasks are the UAV path trajectory and length, devices energy consumption, and communication distance between WSN nodes and UAV. In this paper we consider the “agent” notion as an equivalent for the following: sensor, cluster head (ground-based agents), UAV and mobile sink (aerial agent). In our interpretation, the entire WSN–UAV system is a heterogeneous multi-agent system.

The task of designing procedures for generating paths that optimally fulfill the UAV and WSN system requirements, in terms of distance and energy consumption, while simultaneously respecting collision avoidance is a matter of great interest. There are several criteria that characterize and influence the path planning procedures:-offline versus online, which differentiates between the flight preparation phase or during the actual flight;-coverage versus waypoint passing: the former entails a discovery phase in which features like sensor position are estimated while the latter focuses on the interaction with the WSN;-static versus dynamic: is the path update carried out at runtime or not?

Many works consider various restrictions that affect the path planning procedure. These range from:-geometric: the imposition of artificial limitations in the types of paths, which may result [15,16,17,18];-energy-based: considerations of the energy expenditure and distances within the WSN [19,20], limited buffer capacity for the WSN’s sensors [21];-internal dynamics: fixed-wing UAVs have a higher cruising speed and longer operation time whereas rotor UAVs are more flexible but have less autonomy [22,23,24];-communication: signal attenuation due to obstacle occlusion [25], minimum communication time with the cluster head [20].

In general, the path generation is a complex procedure stemming from the nonlinear nature of the UAV dynamics and of the various operational constraints, which include: waypoint passing, line-of-sight communication, collision avoidance, etc. All these elements lead, in its more general form, to a nonlinear (in cost and constraints) constrained optimization problem, which is often impractical to solve [26,27].

Noteworthy, the schemes encountered consider mostly a single UAV operation scenario. Those which have multiple UAV usually partition the tasks a priori such that each of the UAVs functions independently of the others [27,28].

The rest of the article has been structured as follows. Section 2 defines a specific context into which our contribution can be framed. This is mostly related to a significant increase in the interest of the research community in studying and developing UAV, WSN and IOT systems. Our primary analysis is based around keyword searches in the reference SCOPUS and Web of Science (WOS) databases and we detail the arguments for the articles that we have surveyed for this review. Section 3 is focused on the advantages and specific design space of collaborative UAV–WSN operation. One of the main topics that is handled in this section relate to designing efficient UAV paths and tracking in mission scenarios that involve WSN data collection. We also discuss the key challenges and current approaches for the data acquisition and data processing, which can leverage the on-board computing resources of the network nodes while reducing communication bottlenecks inside the collaborative system. Due to the varied radio and networking protocols that are usually deployed, we handle data communication as a independent topic in Section 4. This details the radio interfaces that are used at both the WSN and UAV levels as well as the components, both logical and physical, that enable interoperability in a hierarchical fashion, from low power/low data rate to high power/high data rate. Section 5 handles implementation aspects of integrated UAV–WSN systems with a main emphasis on optimization algorithms and heuristics that allow improvements in system performance for data collection, relaying and extended autonomous operation through energy efficiency. The core applications that we survey as representative use cases for collaborative large-scale monitoring systems are in agriculture, environmental monitoring and disaster management. These are discussed in Section 6. The article concludes with discussion and conclusion sections that observe the current trends and main areas for a research agenda in this field.

For readability, each topical section includes a synthesis table with the main relevant articles that were analyzed, highlighting the key objectives and approaches to the reader. To save space and to better track the work content, Appendix A gives a complete list of the acronyms and symbols used throughout this review article.

## 2. Materials and Methods

Although the works that presented separately WSNs and UAVs are older and their respective research topics are well-established in the literature [29], the study of collaborative integration between UAV and WSN is relatively recent (mostly after 2004). One of the first reference papers addressing an integrated WSN–UAV approach is [30], where an UAV is used for WSN node deployment. In [31], considering the drop in the cost of the drones, a multi-UAV system is proposed for collecting data from ground sensors. To this end, mobile software agents were used for intelligent communication, data collection and processing.

In order to better frame the opportunity for the current work we have initially performed a keyword search across the SCOPUS and WOS databases in order to have a better picture on the main topics of this survey. We considered a period of 12 years between 2007 and 2018. The search was split between individual keywords and combinations of keywords using “and” and “or” connectors, while searching the title, abstract and keywords of the respective original articles indexed in the databases and reporting the yearly publication counts. For the individual keywords we searched for the following keywords: UAV, WSN and IoT. Figure 2 and Figure 3 show the individual results for Scopus and WOS databases respectively. While we observed a linear increase in the interest from the scientific community for the UAV topic, for WSN there was an increase until around 2012 followed by a relative stagnation beyond this point (or an even slight decrease in terms of WOS data). This was more than compensated by the exponential increase in the interest for IoT and can be explained through the fact that many current IoT systems rely on wireless sensor networks as key building blocks for their implementation. Among the two studies databases the trends in publication counts over the years were similar, with Scopus providing an absolute larger number of references when compared to WOS as well as timelier indexing of the respective contributions.

For the combined study, we searched for the combinations: UAV and WSN, UAV and IoT, UAV and satellite and UAV or WSN, UAV or IoT. In the “or” case, as expected we saw the cumulative effect of both areas of contributions in terms of publication numbers over the twelve years. We might therefore conclude that these topics represent large and growing areas of study. At their intersection, in the “and” case, by adding up the two combined keywords we highlighted an increasing interest in studying and developing hybrid large scale monitoring systems based on both UAVs and WSN/IoT ground solutions. This serves as a proper justification for the current survey. In addition to the previously defined keywords, for the “and” study we also included “satellite” in conjunction with UAV and IoT. We argued that there was also an important number of publications, especially in the telecommunications community where a satellite was seen as the third link in hybrid large scale monitoring systems, which complements as a redundant backhaul the low power radio network of the WSN/IoT and the GSM or other type of long-range communication links on-board the UAV platforms. We also performed a more restrictive SCOPUS search for the combined UAV–WSN–IoT keywords. Five articles [32,33,34,35,36] were identified and referenced, which covered all these topics in areas relating to enabling improved data collection, softwarization of large scale monitoring systems and agriculture IoT architectures and deployments.

Finally, the combined results of this literature meta-review are illustrated in Figure 4.

For this review 985 papers were found from different databases like: WOS, Scopus, IEEE Xplore and PubMed, of which we selected and researched 121 for this review. The main criteria for selecting or rejecting a paper was addressing of a UAV–WSN integrated system. We expect for identified early papers covering this topic, the authors focused on related work published in the last 5 years to better capture the technical novelties proposed in the literature. Under these considerations, papers using both UAV and WSN but with no collaborative functions were not considered relevant for the scope of the paper. Papers were classified according to the main and secondary topics addressed: architecture, path planning, data acquisition and processing, communication, simulation, real implementation or specific application. Relevant papers were first selected for each topic based on the work visibility and impact of contributions (publishing in high-rank conferences and journals and number of citations), based on the technical novelty and relevance of the work and considering this survey’s authors’ perspective on the domain and topics addressed. Most relevant papers were detailed in each article section. The other less relevant papers, which also included useful experimental results or approaches, were included only in the article’s tables.

## 3. Collaborative Operation in UAV–WSN Applications

### 3.1. Collaboration and Intelligence in the UAV–WSN System

One of the first collaborative UAV–WSN applications was the location of sensors on the ground. Thus, in [30] the AVATAR autonomous helicopter is used for WSN node deployment. The setup allowed establishing communication between WSN nodes and a helicopter as a feedback on the deployment algorithm. WSN modes were Mica Motes and the communication between UAV and WSN used radio signals at 915.5 MHz and achieved a maximum distance of 13 m.

More recently, UAV–WSN architectures have been using aerial vehicles as one of three types of actors:-for node deployment [30] in applications with low on-site accessibility or dangerous for human operation;-as actuators, as in [37] where drones are used for spraying pesticides;-as relays for receiving and retransmitting a signal, as in [38], between CH and GDT, in [39,40] as message ferrying in sparse networks, in [41] as a mobile node carrying sensing equipment or in [42] to overcome faults of the sensing network;-as mobile sinks, this being the most widely used role integrated with either small or large scale WSNs. In this case, two data acquisition modes were identified:○clustered, when data is sent to a local CH;○direct communication, where the UAV collects data from each sensor node.

In [43] the authors provide a model depending on weight, wing area, air density and energy required for transmission for evaluating the energy consumption for a UAV. They compare the clustered and direct communication approaches and show the energy consumption is lower in the clustered design. UAV were used for the dynamic clustering of ground sensor nodes and cluster head selection mechanism in a randomly deployment of sensors [1].

Due to the development of new technologies, the trade-off between storing and transmitting WSN ground level data is, currently, a challenge [44]. The UAV acts as a mobile CH for the network with a novel scenario in which it harbors an energy harvesting module and wireless recharging for the ground nodes. Energy harvesting is based on RF signal energy extraction where part of the received signal is used for this and part for standard data collection while hovering over a sensor node island. Based on information collected from the nodes, the UAV schedules appropriately the battery charging and data gathering. Visiting the nodes is based on a variant of the travelling salesman problem (TSP) with time windows: shortest travelling time trajectory (STTT). The model used for the battery level of a WSN node is a birth–death process based on the observed activity rates in terms of data sensing and transmissions.

### 3.2. Satellite Information

Satellite connection offers two major advantages: system localization (both UAV and WSN) and data communication support. For localization the system elements (especially the UAVs) are equipped with GPS. One of the first notable papers introducing the use of a UAV equipped with a GPS as a beacon node was [45]. In this case no a priori knowledge on the WSN node location is required. For WSN node localization the GPS information is combined with the RSSI signal strength of RF communication between the UAV and the node [1]. The position is represented as a probability distribution function. Experimental results prove that by increasing the distance between node and WSN the standard deviation of the RSSI signal decreases [45]. Therefore, the positions are less sensitive to measurement uncertainties and the signal does not need to be filtered.

### 3.3. UAV Path Generation and Tracking

In the context of WSNs, the main motivation of having an UAV (or team of UAVs) as mobile sink(s) is to prolong the sensors’ lifetime (by canceling their need to communicate with a base station [20]) and to reduce operational costs (canceling the need of direct human supervision [25]). In addition, cases where widely dispersed parcels must be traveled preclude direct sensor communication and unavoidably require UAVs [22,23].

Thus, gathering and exchanging information with the WSN implicitly translates into the design and subsequent tracking of a path (or paths in the multi-UAV case [28]). This is hence the main justification for our interest in path planning and the ancillary mechanisms: whereas the WSN–UAV system may have different objectives and constraints, ultimately these translate into limitations and restrictions of the mobile agent’s path.

There are several criteria that characterize and influence the path planning procedures.

(i) Offline versus online planning. Most works assume a known environment (known domain, areas of interest, interdicted regions, etc.), which permits to solve the motion planning problem offline. A varying environment or changing mission parameters may lead to an online computation of the path. This is generally avoided as it introduces elements of risk, which are unacceptable in practical applications. Still, path updates are sometimes employed, either due to alarms signaled by a collision avoidance mechanism [46] or due to mission updates [24].

(ii) Coverage versus waypoint passing. Some works consider a discovery phase in which the deployed sensors’ position has to be estimated or the terrain has to be mapped through a preliminary pass. Such missions cover the entire area of interest [47] through either random or deterministic scanning. The former may be considered, e.g., in military applications where random routes are selected to avoid prediction of future actions. Realistic implementations combine both types of paths even when the sensors are deployed deterministically (e.g., in a grid) [48]. Furthermore, obstacles or other features of interest may not be stored in the static map available at the mission’s start.

(iii) Static versus dynamic path planning. Whereas the path generation procedure is mostly static (decided at the ground level prior to the flight), novel information is sometimes used to update the path. As noted in [19], predefined routes are dangerous as any un-modeled disturbance can greatly impact the performance. Thus, more advanced schemes account for obstacle and collision avoidance (i.e., sense and avoidance strategies [46,49]) and even for online path reconfiguration [24]. E.g., the authors adjust the path by considering both static requirements (accounting for borders as it has to stay in a safe lane) and dynamics requirements (counteracting wind gusts and avoiding regions, which are already sprayed, based on information sent by the ground sensors).

#### 3.3.1. Limitations in Path Generation and Tracking

Separately, we might consider the various restrictions that affect the path planning procedure. These range from geometric, energy-based, internal dynamics to communication restrictions (or a combination of them).

(i) Geometric restrictions. Due to the inherent difficulty in solving a generic motion planning procedure some authors impose artificial limitations in the types of paths that may result (thus simplifying the planning procedure). Most commonly, this means limiting the possible orientations of the mobile agent (the UAV) along its path. In [15] the authors consider data collection in a harsh-undulating terrain with the base station located far from the sensing region and the way-point selection is simplified by allowing only forward and axial flight (movements across a rectangle-shaped region where the cluster heads are positioned in rows and columns). A similar approach is followed in [16], which proposes a grid division algorithm. [17] forces strip-based and zig-zag paths. Authors in [49] go further by imposing the avoidance of predefined regions. Somewhat differently [18] groups the sensors in concentric shells and maps a spiral path passing between them.

If the monitoring zones have obstacles or interdicted regions, then they must be circumvented on the basis of a predetermined trajectory, if the obstacles are fixed [1], or recalculated, if the obstacles are not known in advance or are mobile [46].

(ii) Energy-based restrictions. In [19], a mobile sink routing algorithm is used to compute the energy expenditure and distances within the WSN (further used in the generation of an optimal path for a single mobile sink). [20] proceeds similarly by limiting energy consumption (both due to trajectory and communication requirements). Noteworthy, [20] considers multiple GDTs to ensure telemetry with the UAV and [21] assumes limited buffer capacity for the WSN’s sensors and solves simultaneously the path generation problem and gives the multi-hop rules within the WSN.

The limitations of the mobile agent may also influence the network’s structure: too many and too far away clusters may be impossible to reach in a single flight by the UAV (due to fuel limitations) [15]. This leads to formulations where, for a given flight time and/or available energy budget, the optimal path which passes through the largest number of sensors is computed [17].

Other performance criteria may be employed, e.g., [50] considers a “total flying score”, which combines multiple factors in the path planning procedure.

(iii) Internal dynamics restrictions. In many small to medium scale applications (e.g., in precision agriculture—sensing weather and soil conditions; pest and weeds management; animal attacks and crop morphology) [22,23] the UAVs employed are quad (or multi) rotor UAVs. Large scale applications (e.g., pesticide spraying [24], crop monitoring, etc.) are mostly handled by fixed-wing UAVs. The former can track complex trajectories (where both position and yaw angle are controllable outputs) and the later are usually limited to a constant-height flight plane (where the velocity and heading angle are controlled). Overall, fixed-wing UAVs have higher cruising speed and longer operation time whereas rotor UAVs are more flexible and require less maintenance but have lower autonomy.

Some common drawbacks affect all trajectory procedures. Foremost among these are wind gusts [22] and position measurement errors (e.g., location through GPS is imprecise; dead reckoning is time-sensitive).

(iv) Communication restrictions. When the field contains many obstacles, the communication link may be weakened or lost as a result of signal attenuation [25]. Almost all the surveyed papers consider a necessary time for data transmission at the cluster head; they mostly dispense with it by assuming VTOL (vertical takeoff and landing) UAVs (mini-helicopters or, more commonly, quadcopters), which can hover indefinitely. To ensure that the mobile agent is within the cluster head communication range until all the information from that fixed agent is retrieved imposes additional restrictions. These may be implemented either explicitly by computing the time in which the UAV lies within the communication radius of the cluster head [20] or implicitly by proposing a path guaranteed to spend enough time in the neighborhood (e.g., via loitering [51] or hovering [22]).

#### 3.3.2. Waypoint Selection, Ordering and Passing Through

Computing a trajectory most often reduces to finding an ordered collection of waypoints through whose neighborhood the UAV has to pass [22]. This neglects the internal dynamics of the UAV agent (the trajectory is simply a union of segments that link consecutive waypoints) but is a reasonable assumption in most cases and is justified by the fact that most (if not all) commercially or academically available autopilots (the ensemble of software that ensures path tracking) expects a list of waypoints as input. This simplification becomes less defensible on smaller distances where the UAV agent dynamics become significant (e.g., the turn radius cannot be ignored).

Applying both geometrical/energy-based on one hand and communication restrictions on the other hand is challenging. Thus, many works first consider only geometrical/energy-based restrictions, which give an unordered list of waypoints and later apply communication restrictions to decide in which order should the UAV agent cover the waypoints.

The ordering may be decided online as in [15] where the next way-point is determined by analyzing the information provided by the existing nodes (the closest, still not visited node, is chosen). However, most papers consider variations of the TSP for finding an optimal route among the waypoints [19]. A more complex representation appears in [24], which considers a second pass through (some) of the waypoints if there are still areas uncovered from the first pass.

Noteworthy, the schemes encountered consider mostly a single UAV agent. Those that have multiple UAV (hence a multi-agent UAV subsystem within the larger, heterogenous, multi-agent system of WSN plus UAVs) usually partition the tasks a priori such that each of the UAVs functions independently of the others [27].

Simply passing through a way-point or through its neighborhood is almost never sufficient. Most applications require a hovering time (such that there is enough time to gather the required information). In such cases, the distinction between multi-rotor and fixed-wing UAVs becomes significant: the former can hover easily and for an arbitrary amount of time whereas the later have to enter a so-called loitering mode [51] in which they orbit around the current way-point (the minimum radius depends on the UAV characteristics).

Some papers consider an explicit communication range when imposing hovering conditions [17,20].

In conclusion, the path generation procedure is affected by multiple restrictions and must consider several aspects: -properties of the WSN: is sensor localization required or available [1]; the size of the WSN area; data collection from sensors, sensors activity monitoring relay function [52] and energy loading by radiation [44];-characteristics of the used UAV agent: available energy, type (rotary or fixed wing); type of antenna and speed;-characteristics of the environment: if there are obstacles or the application is in open field; the availability of ground base-stations along the entire path.

Table 1 summarizes which of these topics were addressed simultaneously in different papers.

#### 3.3.3. Computational Aspects

The path generation is a complex procedure due to the nonlinear nature of the UAV dynamics and the various operational constraints (waypoint passing, line-of-sight communication, collision avoidance, etc.). All these elements lead, in its more general form, to a nonlinear (in cost and constraints) constrained optimization problem, which is often impractical to solve. Even a relatively simple requirement as ensuring hovering at the waypoint leads to a mixed integer nonlinear problem (MINLP) formulation [15].

The usual solution is to simplify the motion planning procedure. This may happen either at a conceptual level by imposing a limitation that lead to a manageable formulation or by solving the ‘hard’ problem heuristically through algorithms providing sub-optimal but fast solutions. The former means considering limitations on the types of paths taken by an UAV (spirals [18]; zig-zags [17], etc.). The later approach is concerned with problem reformulations and the application of heuristic algorithms.

Even when mixed-integer (MI) formulations appear explicitly, they are often solved heuristically as in [28], which ensures obstacle avoidance through a MI formulation solved by a genetic algorithm. Xu et al. [20] also employs binary variables to check whether the UAV is in contact with a given sensor and relaxes the formulation through time allocation tactics and pre-scheduled GDT usage. When obstacle avoidance is considered, rapidly exploring random trees (RRT) and optimal RRT (RRT*) algorithms are often the preferred method [46].

Among the heuristic algorithms used we may mention: iterative genetic algorithms (GA) [22]; ant colony optimization (ACO) [53] for AMR (automatic meter reading); fast path planning with rules (FPPWR) [16,18]; potential field methods [54] and greedy algorithms [17]. Note that all such heuristic methods have (in varying degrees and with varying compromises) common qualities and defects. On the one side they provide quick and near-optimal solutions (sometimes orders of magnitude faster than exact optimizations) but on the other side lack robustness guarantees and may become stuck into local minima (and thus, fail in solving optimally the problem).

It is important to note that only a few of the surveyed papers validate their results on experimental benchmarks [47,48,50]. Most of them limit themselves to simulated environments instead. Moreover, most papers do not consider obstacle avoidance issues. None of the reviewed papers addressed simultaneously path design with obstacle avoidance in a real environment.

Table 1 summarizes the important aspects of paths generation in UAV–WSN systems.

### 3.4. Data Acquisition

As one of the core functions of the integrated UAV–WSN monitoring systems, data acquisition is concerned with the correct and timely collection of ground level measurements and their propagation through the network hierarchy. This is implemented in various ways, mainly by using the UAV platform as a collection agent for the lower level sensors. Key technical challenges for UAV–WSN data acquisition concern the aggregation at the ground clusters and the frequency and route for the data collection based on the design decisions at the system level and the optimization problem formulation.

The main advantage for using UAVs as mobile sinks is that WSN nodes can reduce energy consumption by reducing the need for networking overhead at the local level with functions such as discovery, connectivity and maintenance of routing tables being maintained by the UAV/mobile sink, which periodically updates the cluster nodes with the required information for packet transmissions. In many cases the UAVs were considered as relays for receiving signals from CHs or other UAVs and then retransmitting the signal to the GCS. The most important requirements in data acquisition in the integrated UAV–WSN systems are the energy efficiency (primarily to maintain the energy capacity of sensory nodes and, secondly, to increase the UAV coverage area), the accuracy and the time of data acquisition.

Performance analysis for WSN with UAV support is carried out by [63]. The UAV play the role of mobile sink across clusters of ground sensor networks. A spectrum allocation scheme is modeled in order to achieve robust communication with concurrent transmissions from the WSN and/or multiple UAVs. The data transmission capacity at the node and network levels is analyzed. Monte Carlo simulations are performed to compare theoretical per-node capacity and UAV working time models to varying parameters. The per-node capacity is defined in terms of the interval time between the collection of accumulated ground data. The multi-UAV scenario assumes a returning path to avoid collisions of packet transmissions between adjacent UAVs. The routing of the UAV is based on pre-defined ground cells, which hosts groups of sensor nodes and the optimal number of cells can be identified that maximize the per-node capacity of the WSN thus improving the overall data collection procedure.

The usage of UAV platforms to support mobile devices is underlined in [17] as a promising solution for infrastructure less integration. In this context infrastructure, less integration is defined such that the ground sensors are deployed independently and they only communicate directly with the UAV, without local interactions that requires a pre-established network structure, cluster heads or additional base stations. The main approach aims to minimize overall system energy consumption (mobile devices + UAV) under typical real-world constraints for resource allocation, UAV flying path and computation offloading. The solution to the non-convex formulated optimization problem is presented by means of decomposition into manageable subproblems. The computational status at the mobile device level is analyzed by simulations in conjunction with the UAV path planning for task offloading on a slot by slot basis. The metrics against which the optimization scheme is analyzed are related to the average energy consumption of the mobile devices and UAV per slot, only the average energy consumption of the mobile devices or only of the UAVs, expressed under predetermined simulation conditions. Several scenarios are studies where the UAV flying path is closer to the mobile devices to provide high channel quality and computational offloading resources thereby reducing the local task queue length to be processed.

The impact of UAV mobility patterns on the quality of the data collection for ground sensor nodes is analyzed by [64]. Initial sweep flights at different altitude levels are performed over the area of interest to discover and enable the sensor network. Four mobility patterns are subsequently defined as: tractor, angular, square and circular patterns. Simulations based on these are implemented in OMNET++ and MiXiM frameworks for IEEE 802.15.4 compliant wireless sensor nodes. The number of clusters formed, and implicitly associated CHs are reported under static ground simulation conditions and variation of the UAV altitude. The metrics used for evaluation include the overall coverage by the UAV, time efficiency and utilization, time versus coverage efficiency. The results can be used for preselection of a mobility pattern for the UAV upon mission configuration while selecting the ground clustering mechanisms based on this.

The main use cases for symbiosis between UAV and wireless networks relevant to the study are related to providing IoT uplink connections in an energy efficient manner with reliability as well as support for disseminating information and connectivity enhancement [65]. The UAV is able to access the WSN/IoT ground network directly or by means of a machine-to-machine (M2M) dedicated gateway and further integrate with new 5G IoT services in various industry verticals. The types of communication protocols that can support such hybrid systems are classified into: proactive protocols, reactive protocols and geographic protocols where the design trade-offs between metrics such as latency, coverage, energy-efficiency, etc. are known and established beforehand in the system design phase. Security and privacy also play a critical role in ensuring that proper safeguard for in-network information collection is in place.

The authors of [66] focus on the clustering methods for the ground sensor nodes in a single UAV scenario for environmental monitoring and data gathering. The energy model for the sensor nodes is composed of the energy needed for packet reporting and for packet forwarding in the case of mesh networks where each node can serve as a router for neighbor packets towards the sink. UAV path planning uses the 2-opt heuristic, which is a local search algorithm to solve TSP for visiting the WSN clusters. The evaluation consists of a 100 m × 100 m grid with 200 randomly deployed nodes according to the random Poisson point process. Nodes are modeled with cc2420 family radio interfaces and for the UAV the characteristics of the DJI Phantom 3 drone are used. Optimization problem is solved using the CPLEX solver. Results plot the energy consumption of the UAV and of the sensors versus the maximum number of hops in the network.

Optimizing for UAV flight time reduction in data collection missions that support ground sensor networks is discussed by [67]. The scenario assumes controlled deployment of the WSN in a straight line with cruising or hovering behavior from the UAV. The aim is to achieve optimal, non-overlapping, data collection intervals, UAV speed and sensor transmit power by using a dynamic programming (DP) approach. The main finding highlights how the UAV speed should be proportional to the energy levels of the sensors and the inter-sensor distance. Several scenarios are defined for evaluation under randomness assumptions for data requirements, energy and localization of the ground nodes. While cruising, it has been found that even if the data rate of the nodes decreases, the reduction in the mission time is able to compensate for this in comparison to hovering. This is applicable to low amounts of data, whereas, beyond a certain threshold, the hovering approach is preferable.

The problem is framed as an aerial data collection problem (ADCP) with increased complexity given the fact that a set of UAVs is tasked to collect information from mobile ground sensor nodes [68]. ADCP is solved optimally for small instances using mixed-integer linear programming. Collected data is relayed back to the central base station using multi-hop aerial communication links over the set of UAVs in a multi-tier network architecture. Simulations are carried out by means of a custom implementation using the JGraphT library and IBM CPLEX solver for the optimization part. Computational test results and scalability results are presented with various UAV and WSN sizes and deployments. Suitability of the approach for real wildlife application is foreseen. A heuristic pricing scheme shows promise to solve efficiently the ADCP formulation in the defined 3D-positions of the targets while accounting for mobility and connectivity variations.

#### Data Mule Scenario for Data Acquisition

Data mulling concerns the effective creation of a communication link between disparate network subsets by usage of a mobile node. This helps create a data bridge among clusters and earlier works discussed the implementation of data mules by means of mobile robots. In [69] the optimization of the shortest data mule path is carried out by an improved clustering-based genetic algorithm, which enhances the travelling salesman problem with neighborhoods. For larger geographical distances where a ground robot solution is unfeasible, UAV-based systems have been integrated [70]. Although in most of cases the communication between WSN and UAV is unidirectional, authors from [71] proposed dual stack single radio architecture, which allows a UAV to communicate in a bidirectional manner with a WSN (UAV-to-WSN and UAV-to-Sink). It is used a single radio transmitter both for collecting data from the WSN and for sending telemetry data to a Base Station.

The GrubiX Wireless network simulator has been used to comparatively present the results of two optimization heuristics for path planning, namely the Euclidean travelling salesman problem (ETSP) and a modified version with cluster selection rounds. For radio link modeling the IEEE 802.11b standard has been selected in ideal propagation medium with 50 m communication range. The scenario in which the UAV is used as a data acquisition mule among clusters of separated ground sensors is handled as a specific use case of distributed data acquisition in collaborative UAV–WSN systems. The data mule scenario is graphically depicted in Figure 5. The interaction is handled at the CH level (blue node) by using an interoperable radio interface on-board the UAV. This is specified according to the data type and size that is to be collected, while considering the flight characteristics of the UAV. Optional satellite-enabled communication support can be integrated in order to provide synchronization services to multiple-UAV teams operating in remote areas.

### 3.5. Data Processing

In many application scenarios, relaying the raw collected data from the local sensors to the central aggregation point of the integrated system poses a considerable burden on the computing and communication resources. In these cases, some form of processing is necessary to compress, reduce or intelligently extract information from the sensor measurements. The analyzed articles propose a wide range of methods: from basic processing (min/max thresholding, averaging and basic statistical features) to higher level distributed approaches such as fog computing.

The authors of [72] point out the key areas of active research for data aggregation within IoT systems of which WSNs are a core building block. The potential for increasing the energy efficiency of the network through clustering mechanisms for data aggregation is acknowledged alongside tree-based and centralized approaches. The respective trade-offs for each family of methods are identified in terms of advantages and drawbacks concerning energy, traffic load, accuracy, security, scalability and fault tolerance. Depending on the constraints of the application of the UAV–WSN collaborative network, a suitable aggregation method will be employed. This is determined by the streaming or event-based nature of data generation, as well as latency and energy constraints. Clustering approaches are generally more robust than centralized schemes, which have a single point of failure, with the added complexity of managing and organizing clustering hierarchies.

Fog computing supporting and UAV–WSN deployments are discussed by [73] in which the UAV plays the role of the fog node and provides services to the hierarchical ground sensor network. The services include data collection and relaying, local processing and cloud interfacing. Several relevant use cases are identified ranging from ground pipeline monitoring and control to underwater monitoring, large scale emergency response and military deployments. An evaluation is carried out by comparing the performance of local and cloud service calls in various network configurations. The resource discovery and integration are also enabled through the UAV fog node with additional benefits stemming from the inherent mobility of the UAV in assisting the ground level devices. Security functions are also available on the fog level with application-based protection levels.

The fog computing paradigm applied to UAV-based large-scale monitoring systems is analyzed, which assumes that the path planning, flight coordination and task planning are carried out locally at the fog coordinator level [74]. This is reflected in comparison to the usage of a cloud platform, which potentially offers more computing resources and knowledge with increased data and command latency. A system model is provided for the throughput, power consumption and latency analysis with multimedia data over various mobile communication technologies such as GSM, UMTS (Universal Mobile Telecommunications Service) and HSPA(+) (High Speed Packet Access). A single UAV is denoted as fog coordinator and it decides on the trade-off between local processing and cloud offloading of the data streams and planning tasks, based on a genetic algorithm optimization method with implementation in MATLAB. The end-user application can be considered as an UAV–WSN collaborative monitoring scenario in which the fog computing approach is deployed at the higher UAV ground control station (GCS) level.

Cloud-based support infrastructure for UAV–WSN mixed data collection platforms are introduced. It is argued that the increased availability of cloud resources can contribute to the improvement in the flying parameters and information extraction from the WSN data [50]. Emerging ground level events are classified according to predefined priority levels and the flying sequence is determined by considering several correlation factors at the node and cluster levels. Results are evaluated both in simulation and through experiments on an integrated testbed using a quadcopter UAV and CC2530 2.4 GHz radio sensor nodes with on-board GPS capabilities. The proposed method, denoted as cloud-assisted and weight event data collection (CWC), is compared (in a simulation framework) to a full collection method (FCM), event collection method (ECA) and event collection method with priority (ECP). CWC performs well in terms of reducing the flight time and distance as well as assuring a 97% data integrity compared to 60%, 63% and 26% for ECA, ECP and FCM respectively. Data integrity is defined in terms of the sum of the total received messages compared to the total messages.

UAV-enabled WSN data aggregation is described by [59] in which the UAV platforms act as data mules between the CHs of the network. The routing optimization is based on genetic algorithms (GA). Energy consumption of data transmission based on a standard energy-per-bit transmission model is used as minimization objective. The OmNET++ simulation environment is also used to deploy a modeling scenario of the proposed approach. The evaluation is with reference to three other protocols: centre-based, greedy-based and clustering-based genetic algorithm (CBGA) and achieves an improvement in terms of system-wide energy consumption of 1–28.4%. This is due to the fact that the cluster member energy expenditure becomes an explicit optimization objective. Shorter data update periods are also listed as a secondary benefit of the approach.

The challenge of information fusion for collected sensor data from both fixed and mobile devices is illustrated in [75]. The application focuses on deep supervised learning in transportation systems by integrating GPS, GNSS and accelerometer readings with remote sensed imagery. The solution covers both the management of the large data quantities as well as analytics for pre-processing and feature extraction. Evaluation analysis is performed for a transportation mode recognition task when various data sources are considered, and the best accuracy is at 97% for the combination of GPS + accelerometer + GIS data. The work is relevant also to the large-scale hybrid UAV–WSN monitoring systems for distributing the pre-processing and feature extraction modules across the nodes of the network and re-tasking in accordance to the mission objectives.

At the ground sensor level, distributed agreement schemes such as consensus algorithms can contribute toward intelligent data reduction and reducing the burden on the upper UAV layers for processing and transmissions [76]. For event detection where the integrated system has to assure the timely observation of mission-specific events, local analysis can improve the overall detection latency. In this manner each of the ground sensors periodically communicates their findings with neighboring nodes in order to reach a joint decision, such as the event present: yes or no, in as few as possible communication instances as possible. The weighting scheme accounts for the quality of the data as well as the reliability of the information provided by each node. Implementing consensus local decision algorithms in large scale UAV–WSN monitoring systems can help increase the reliability and robustness of the overall system.

In Table 2 some characteristics of the joint data acquisition and processing aspects of collaborative UAV–WSN systems are presented.

## 4. Data Communication

Data communication performs the critical integration function of UAV–WSN large-scale monitoring systems. The various protocols, standards and physical interfaces have to be interoperable and tuned for adjusting the system performance to the (dynamic) mission objectives. As previously illustrated in Figure 1, these can range from low-power and low-range radio communication at the ground level, to higher throughput radio links for data streaming, complemented by satellite communication. Though implementation it is mostly wireless also, wired links can serve as a data communication backhaul for the integrated system.

### 4.1. Requirements and Protocols

As a preliminary step in operational deployment of integrated UAV–WSN monitoring systems, the deployment strategy of the ground sensor nodes can be considered [83]. DeVForce-AP algorithm is presented, which allows the elimination of coverage gaps in the ground network by means of Delaunay triangulation together with DeVForce for maintaining network connectivity. Parameter adjustment is realized to account for environmental factors. The method evaluation is presented in relation to random deployment, traditional Delaunay triangulation and virtual force deployment. The new method presents an improvement in the data transmission success rate, the number of alive sensor nodes and the number of available neighbor nodes. It is also able to maintain coverage ratio and increased network lifetime under varying environment conditions and obstacles.

The trade-off between storing and transmitting WSN ground level data is discussed by [44]. The UAV acts as a mobile base station for the network with a novel scenario in which it harbors an energy harvesting module and wireless recharging for the ground nodes. Energy harvesting is based on RF signal energy extraction where part of the received signal is used for this and part for standard data collection while hovering over a CH. Based on information collected from the nodes, the UAV schedules appropriately the battery charging and data gathering. Visiting the nodes is based on a variant of the travelling salesman problem (TSP) with time windows: shortest travelling time trajectory (STTT). The model used for the battery level of a WSN node is a birth–death process based on the observed activity rates in terms of data sensing and transmissions.

The system designed for an IoT surveillance application is presented by [84] and is based on visual ground sensor network with security cameras, patrolling drones and unmanned ground vehicles. The integration and the data and command flows are detailed using open IoT protocols such as MQTT and publish–subscribe data architectures. Performance of the combined system is evaluated based on object detection time under varying number of frames. The edge computing approach offers a considerable improvement over conventional cloud implementation in terms of measured detection latency. The scalability test shows how the integrated surveillance system performs under increased device numbers. The system provides a blueprint for our reference UAV–WSN large-scale monitoring scenario by implementing a decentralized solution based on the edge and mist computing paradigms.

Paper [85] focuses on the hierarchical system architecture design for hybrid UAV–WSN monitoring systems. It includes a central gateway system that coordinates the UAV platforms within the aerial level, followed by the distributed ground sensor systems for monitored parameters data collection on the ground. In addition, a binary sensor level can be also added in tight connection to the WSN and without local processing elements. A clustering mechanism is implemented and evaluated under the Contiki embedded operating system for resource constrained devices, which considers radio link indicators such as received signal strength indicator (RSSI) and link quality indicator (LQI) for selecting the cluster heads. Static overpass UAV paths are simulated to obtain in clustering behavior of the WSN using Rime broadcast and unicast messages from the nodes. An application of the proposed clustering mechanism shows improvement in data collection and distributed agreement for physical measurements.

Authors in [86] present a coordination approach for multiple aerial ad-hoc systems integrated with ground systems in order to bridge various configuration options and network dynamics. Their main contribution considers the application of fuzzy bee colony optimization (fBCO) to implement cognitive relaying within the network. Mapping the network structure onto the optimization problem formulation considers the ground nodes serve as employed bees while the aerial nodes act as scout bees. The proposed model is evaluated against several existing approaches such as path-aware geographical routing (TAG), energy-balancing packet scheduling (EPLA) and greedy selection (GS), with 10 UAVs and 10–50 ground nodes. The metrics used relate to average transfer time, network efficiency, average cognitive overheads and convergence time. It is argued that the new fBCO offers improved scalability, lower transfer times and lower cognitive overheads.

The optimization problem for UAV–WSN data acquisition is discussed in [51] considering the main objectives for cluster head selection, realistic models of bit error rate (BER), node energy and flight time of the UAV, including weather effects. The proposed method is compared with the established LEACH-C (Low-Energy Adaptive Clustering. Hierarchy-centralized) for WSN clustering and data collection. Particle swarm optimization (PSO) is used for the multi-objective method, while prioritizing the average BER over energy consumption and travel time, to assure reliable connectivity. Various cluster sizes are analyzed, and it is found that PSO outperforms LEACH-C in terms of the remaining live nodes after several hundred rounds of simulations. Relative weighting of the objectives depends on particular system and mission objectives as well as practical judgment. As an example, reducing travelling time can be related to the operational cost of the UAV and support infrastructure and limited flight autonomy.

### 4.2. Standardisation and Safety Considerations

One of the most sensitive and restrictive issues of UAV–WSN’s integrated systems is flight safety, in terms of the security of people, land and of airspace objectives. There are several sources providing a comprehensive overview of UAV flight regulations, implementation status and forecast. A reliable source of information regarding UAV development, e.g., the Global Drone Regulations Database [87], is providing the interested researchers with free access to accurate description of applicable regulations in each country and standards aiming to minimize the implication in the civilian air traffic operations. Various studies like [88] compared the documents meant to regulate the UAV operations and concluded that there is no general approach of the conditions, UAV classification or standards. Therefore, reviewing the scientific literature presenting the characteristics considered by various countries enabled the authors to have realistic predictions of future trends. Recently, the EASA established (in February 2019) the regulation, which enables the circulation of UAVs within the EU based on security and on increasing development of the UAV industry [89]. Since there are different sizes of drones (from 10 g to 10 tones), flight levels and path lengths, three categories of UAVs have been identified for which specific flight rules have been established: (a) the ‘open’ category (below 120 m level, 500 m distance—line-of-sight, and does not require a prior authorization), (b) the ‘specific’ category (requires an authorization by the competent national authority) and (c) the ‘certified’ category (requires the certification of the UAS, a licensed remote pilot and an operator approved by the competent authority). According to EASA, the UAVs should be safety integrated into the existing aviation context in a proportionate way.

For long-distance applications, in which UAVs are flying beyond the line-of-sight, a very strict regulatory framework needs to be adopted so the UAV’s tracking system provides proper support for a safe operation of the remote command and control, telemetry communication and datalink transmission. The main challenge for the experts of this sector is to determine how to calculate the spectrum demands that most likely need to be harmonized between several entities that will share the same portions of spectrum while ensuring a safe and secure interconnectivity with terrestrial WSN for complex air-ground hybrid networks.

Some organizations are involved in the implementation of the best strategy towards the usage of the spectrum dedicated to interconnect the UAVS and terrestrial WSNs into hybrid networks and, at the same time, to ensure a safe airspace sharing with aircrafts: International Civil Aviation Organization (ICAO), EASA, Joint Authorities for Rulemaking on Unmanned Systems (JARUS), International Telecommunications Union (ITU), etc.

In multi agent UAV–WSN applications, communications between agents (UAV–UAV, UAV–WSN and WSN–WSN are mandatory. Although communication within UAV network or within WSN was not the subject of this paper (only communications between UAV–WSN), some clarifications need to be made regarding UAV–UAV communications. Two solutions can be used for UAV to UAV communication (U2U): direct communication [90] and via GCS (GCS as relay) communication [91]. Direct U2U communication is, on one hand, a difficult problem to solve in terms of frequency regulations versus airspace security and, on the other hand, it is a necessary solution in some applications of a swarm UAVs (examples: collision avoidance, operations of search and rescue). The authors in [90] analyzed the throughput and latency of the UHF (ultra-high frequency) radio link, which interconnect two UAVs. The advantages, disadvantages and important concerns for the U2U communication options are also presented. The indirect communication is presented in more research and application papers. Thus, in [91] the GCS is presented as a relay in a configuration UAV–GCS–UAV system by using commercially available hardware (Wi-Fi) components combined with customized software. In [92] UAV to UAV communication for smart agriculture monitoring is presented using the ad-hoc WiFi infrastructure. The first UAV serves as a relay between the second UAV and GCS, improving the UAV communication in beyond line-of-sight and cross-obstacle operations. The authors present experimental results for various conditions and configurations while implementing the OLSR (optimized link state routing protocol) protocol for increasing the quality of service of the network links. Paper [93] extends this study using RTSP and RTP protocols and in-depth packet and bandwidth analysis for both the video processing pipeline and control functions. The reference UAV type is a quad copter AR drone. The average bandwidth for the UAV system is measured at 15.4 Mbps compared to a pc-only system at 10.91 Mbps. Georeferencing the target area enables precise application of herbicides leading to improved cost-effective productivity. Reference [94] provides a practical assessment of 2.4 GHz and 5.8 GHz UAV communication in emergency response settings. A dedicated system infrastructure is illustrated with GPS, embedded PCs and software components that support the latency constrained operations.

A summary of the UAV–WSN communication characteristics is given in the Table 3.

## 5. Integrated UAV–WSN System Implementation

There have been observed many options for system implementation over the recent years of development. Many research groups first validate their conceptual approaches using simulation and emulation environments where the main components and their interactions are modeled in software. Going from model to real world implementation, the ground sensor node platforms are selected based on their on-board sensors, computing capabilities and communication interfaces. The main options for the UAV platform related to its type: fixed/rotary wing, autonomy, communication interface and other complex sensing systems, which are not suitable for ground implementation, such as high resolution and special application imaging.

Early work in the field of integrated robotic aerial platforms and ground based sensor networks for security applications is presented in [31]. The data processing is organized on three hierarchical levels ranging from target detection, localization and recognition. The ad-hoc nature of the ground sensor network is emphasized as well as the application of intelligent software agents that are able to collaborate for joint mission objectives. The decreasing cost of the hardware platforms also needed to achieve such a system is underlined as an element driving the development of pervasive sensing in large scale monitoring. Adaptive parameterization at both the UAV and the WSN levels enables the system to be robust to environment or mission changes while ensuring proper quality of service with energy efficiency and communication reliability. The combination of low-cost sensors on the ground and high-performance, high-cost sensors on-board UAV sensors allows for data fusion algorithms that can extract relevant information in an online fashion.

Low latency operation requirements and dynamic reconfigurable network topology are identified as key challenges in the design of suitable architectures and algorithms for UAV–WSN joint cooperation [99]. Low-latency routing algorithm (LLRA) is proposed and evaluated for such scenarios in terms of link average delay and packet delivery ratio. It assumes an iterative process to identify the best relay nodes across the network with periodic route updates. This assures connectivity and achieving the minimum transmission delay. The simulation setup includes a layered UAV network with established parameters for directional antenna angle, transmission power, UAV speed and inter-layer separation distance under fixed radio channel assumptions. The interlayer performance of the algorithm is reported favorably with reference to standard routing procedures such as ad hoc on-demand distance vector (AODV) and greedy perimeter stateless routing (GPSR).

Bio-inspired optimization heuristics are applied to collaborative UAV–WSN networks in order to maximize the value of the gathered information from the ground level while minimizing UAV operational costs [61]. With the ground situation known in terms of sensing cells and flying constraints, the authors define four utility functions for the UAV to maximize, in terms of sensing, energy, time and risk. The solution to the optimization problem is achieved using a combination of genetic algorithms (GA) and ant colony optimization (ACO). The best parametrization of the method is achieved while accounting for an additional metric in terms of value of sensor information (VSI) for planning of the flying route among sensing information gathering (SIG) ground cells. The GA and ACO approach also present significantly better results when compared to a classical A* bio inspired algorithm and a combination of A* and GA in terms of the optimal utility value.

A priority-based frame selection scheme within IEEE 802.11 MAC (Medium Access Control) packets is introduced in [100] to mitigate the effect of packet collisions on the overall UAV-assisted WSN network. This operates by assigning a lower contention window range to the higher priority frames. The proposed model assumes the operational characteristics of a fixed-wing UAV in which frames, which are overpassed by the UAV, receive higher priority thus reducing the risk of a data-less overpass. Three algorithms are defined and analyzed as contributions: priority-based optimized frame collection (POFS), priority-based contention window adjustment scheme (PCWAS) and frame selection-based routing protocol (FSRP). The evaluation presented highlights the performance metrics of the proposed scheme in relation to the conventional IEEE 802.11 CSMA/CA MAC in terms of average throughput, end-to-end delay, PDR. Implementation is carried out in MATLAB for a 300 sqm area with variable number of sensor nodes between 100–1000. By including UAV data mule mobility into the prioritization of the communication protocol improved results can be obtained.

The models of UAV–WSN communication integration and data collection are reviewed in [101] for hierarchical, clustered WSNs where the UAV is tasked at visiting each cluster head in an efficient manner to collect the ground level data. The store and forward model are the reference approach in which non-critical data accumulates and is aggregated at the cluster head to be made available to the visiting UAV. Several critical infrastructure monitoring applications might require real-time data streaming or event-based communication, which implies mechanisms and performance constraints on the UAV relays. A hybrid method is also identified, which can prioritize based on the local processing of the collected data and operate the UAV relays in an on-demand fashion. From the UAV routing perspective, a classification is defined based on its capabilities: from constant speed UAV, to variable and adaptable speed UAV and up to hovering capabilities with either unlimited or maximum service times. These have an important impact on the design of the WSN communication windows, which are available to the network.

The approach from the perspective of a collaborative WSN–mUAV network is described for an efficient utilization of the network energy resources [102]. The firefly optimization algorithm (FFOA) is presented for energy efficient relaying of collected data. Benefits are highlighted in terms of continuous connectivity, better network and node lifetime and improved coverage. The studied scenarios assume a single base station with the UAV role in the system model to relay data efficiently between the field nodes and this base station. The optimization procedure selects the best UAV for this based on attraction value among the set of eligible UAVs in the range of the data sink–base station. For performance evaluation the parameters include the followings: throughput, mean hops, packet delivery ratio (PDR), delays, lifetime, coverage and excessive iterations per segment. One example of the achieved results is that FFOA for collaborative UAV–WSN networks achieves 17.2%, 18.01% and 31.5% better PDR when compared to reference methods EEGA, I-ERIDSR and ERIDSR.

The authors of [103] discuss the issue of optimal wireless sensor network coverage by means of UAV platforms in terms of an optimization problem. This is formulated by means of the travelling salesman problem (TSP) to find the best routing of the UAV for data collection with regard to minimizing the energy needed for data transmission of the sensor nodes. The proposed TSP algorithm outperforms particle swarm optimization (PSO) for various flying heights, speeds and network sizes. The algorithm is implemented in MATLAB and simulations are carried out using the OmNET++ network simulator with 802.11b protocol. The reference results for the tested configuration show a lower than half total energy consumption at the gateway level when deploying BL-TSP as compared to PSO and network forwarding. A multiple UAV test shows how the incremental benefits are lower, in terms of cost function decrease, with the addition of new UAVs.

An emulation platform, which serves for generic IoT scenarios, is described by [104]. Main components include the server-side user interface, the physical hardware nodes and the virtual nodes, as well as a script interpreter to handle their functionalities. The system supports both fixed and mobile nodes and the authors present a reference application using Raspberry Pi cards and WiFi communication for the networking side. For visualization a 2D mode for network analysis is provided alongside a rich 3D environment for more immersive views. Implementation is performed using the Java programming language. Running the system in simulation followed by integration with real embedded hardware makes it suitable also for integrated UAV–WSN systems. The application focus for Smart City deployments could be seen as using the ground nodes for environmental sensing with proper task planning of the UAV accounting for the 3D environment in data collection.

The multiagent-based paradigm for developing a highly abstracted simulation model in IoT systems is discussed in [96]. It aims for event and data-driven networks with flexibility and large variety of nodes in increasing abstraction layers. The FABIoT [105] model parameters at the hardware level include the processing power of the nodes, configuration types, percentage of smart objects (SO), if the nodes are battery-powered or not. The network parameters are the time-to-live (TTL) for messages and the ping frequency for the smart objects. Performance is measured using the mean query time (MQT) and plan success rate (PSR) over a reference system of 50 random network configurations of 40 heterogeneous smart objects. Heterogeneity in this case is defined by a mix of hardware and services. Several test scenarios are modeled to reflect the metric variation under typical network assumptions e.g., performance degradation with a percentage of nodes departing/failing.

Communication systems perspective for the ground level WSN is discussed in [106] when dealing with radio challenges over hybrid MACs for decision fusion. The nodes transmit binary non-coherent decisions to the fusion center and the authors analyze and prove several efficient methods that implement the decision fusion rule while accounting for the limited computing available on the nodes. The proposed hybrid MAC provides advantages for detection performance with group size and under low SNR conditions. By grouping ground sensors around MAC schemes, in the same sensor group non-orthogonal MACs are used. The reference for the optimal fusion is considered the likelihood ration (LR) test while three alternatives are developed in order to mitigate computational impact: weighted energy detector (WED), deflection–coefficient- maximization (DCM) and two-step (TS) rules. Evaluation revolves around the detection of a known parameter in Gaussian noise with zero mean and unit variance and receiver operating characteristics (ROC) for various simulation scenarios for the four methods are presented.

Aspects related to the enabling of power efficient communication in WSN with external UAV support are discussed in [107]. A non-convex mixed-integer optimization problem is formulated to minimize total power consumption of the UAV while at the same time offering performance guarantees to the ground sensor nodes for their transmission rate. Block coordinate descent is used for problem decomposition into manageable sub problems. Satisfying of the sensor node quality of service constraints implies reducing the distance of the UAV to it. The three schemes that provide the evaluation imply a power-efficient scheme, propulsion efficient scheme compared to a reference circular flight scheme, under consideration that the UAV propulsion energy cost is dominant against its transmission energy cost generated by initiating or receiving radio communication from the ground nodes.

In previous work we have presented the design of a collaborative UAV–WSN network for environmental large-scale sensing [1]. The main challenges that have been tackled relate mostly to the cluster-based self-organization of the ground sensor network in conjunction with the discovery and path planning optimization. A cluster head selection procedure has been implemented based on the connectivity and signal strength levels of each of the ground nodes during the UAV initial overpass flight. The optimized paths for improved data collection for the UAVs take into account the constraints of the flight zones such as physical obstacles and no-fly areas. A cluster-rebalancing scheme is also presented in order to relieve and balance the energy and processing burden for the individual ground nodes. A realistic testbed composed of an octocopter platform and TelosB-type sensor nodes is implemented and deployed to validate the simulation results.

A summary of the UAV–WSN communication characteristics is given in the Table 4.

## 6. Applications

When addressing specific applications, most authors use a UAV as a mobile sink in either small or large scale WSN. Papers addressing specific applications usually do not detail data gathering or advanced path planning algorithms, but rather how such architectures respond to specific challenges and how a specific implementation achieved application objectives in a certain use-case. The implemented functions depend on each of the application’s objectives. The main applications of integrated UAV–WSN systems are synthesized in Table 5. More detailed characteristics of applications in agriculture, environment and disaster management are provided in the following subsections.

### 6.1. Agriculture

Traditional WSN topologies are not suitable for monitoring small, strictly delimited, dispersed and isolated parcels, which are commonly found in agriculture applications. WSN nodes might also become unreachable as the vegetation becomes denser and performance is affected by the variable node elevation.

Hovering UAVs were used in real conditions for collecting data from cluster nodes in such applications and in [22] reliability at different flight heights is evaluated. Di Gennaroa et al. [109] and Primicerio [110] proposed the use of micro UAVs equipped with multispectral cameras along with WSN meteorological data to evaluate the correlation between grapes quality and measured values.

In the specific domain of spraying pesticides applications, the following challenges were identified:-Some areas might not have the proper amount of chemicals, while other might have a higher level;-The efficiency of the process is highly influenced by weather conditions;-The chemicals must be spread only inside a predefined boundary.

The authors in [24,37] developed methods for these operations, which use sensor measurement values to change the UAV route. The sensors are deployed in a matrix to be able to apply the proposed algorithm. A chemicals concentration map is designed using the available sensor data.

An UAV–WSN–IoT systems in data-driven agriculture is described in [81]. The UAV flight planning is done in conjunction to ground level data depending on the specific parameters to be collected. Duty cycling is applied as well as a scheme that leverages wind speed and orientation to optimize UAV flight time across the fields. UAV video feeds are compared against measured sensor values for improving inference of specific agriculture events. A prediction model is evaluated for inference of precision parameter maps and compared against nearest neighbor (NN) and inverse distance-based interpolation schemes.

A relevant application in agriculture is described by [122]. A 1.9 m wingspan fixed wing Skywalker UAV is used to remotely collect ground sensor reading from commercial Xbee nodes deployed in the field. The parameters include temperature and humidity from a Rotronic HygtoClip HC2-S sensor. The UAV can be equipped with either a normal camera or an infrared camera and the on-board embedded control platform of choice is a Raspberry Pi. Field trial run results illustrate the autopilot enabled operation of the UAV over the target crop while analyzing communication and data collection performance from the Xbee node. A salient feature of the integrated system is that the UAV includes a small on-board tank for deploying directly fertilizers, herbicides or insecticides based on the output of the decision algorithms allowing for fast response to critical conditions with a maximum payload of 2 kg.

### 6.2. Environment

In the environmental domain the integration of WSN with UAV devices led to the development of new applications, rather than increasing efficiency in existing ones, like in agriculture or disaster management. They cover various topics for air pollution, underwater or animal monitoring.

In marine applications, the challenges are given by the difficult access to sensor nodes dispersed over extremely large areas. Thus, [10] proposes an application where several buoys are deployed in a marine environment, with no fixed positions. They collect environmental data like temperature, wind speed, pressure and humidity using the LoRa communication protocol. Subsequently, a UAV is used as a mobile sink to collect data from these nodes. Tests showed that a transmission range of 4 km could be achieved with a data rate of 5.4 kbps.

An advantage of integrating WSNs and UAVs in air pollution is the capability of obtaining a tridimensional sampling of physical phenomena, allowing further analysis and prediction of weather impact and/or pollutant propagation over large geographical areas. Orestis et al. [112] have proposed the architecture for such an application and used a cube map area to model each pollutant.

In the case of wild animals monitoring, the algorithms chosen for path planning must take into consideration their movement over the monitored area. Authors in [80,113] use the metric of value of information (VoI) to characterize sections of a grid where animal movement was identified as an area where intensive data collection is required. This way, the UAV can be directed with priority towards the areas where animals are detected and afterwards cover the rest of the network. Monitored information includes the picture, sound and odor.

### 6.3. Disaster Management

Disaster management applications are characterized by the requirement of operating in harsh environments, where data needs to be collected from dynamic WSNs with high reliability taking into account the fast reaction requirements. A broad study of UAV solutions for disaster management is carried out [115] while considering ground sensor networks that are deployed on the premises and are able to leverage efficiently the aerial support. The design accounts for four stages of disasters ranging from disaster preparedness, assessment, response and recovery. In the first case the system must be optimized for data acquisition and collection while using UAVs as a data mule for relaying in a reliable manner the information of interest back to a central disaster management entity. In the second case, fixed-wing UAVs are employed to scan the affected area and identify target zones for more precise evaluation by means of more maneuverable rotary wing platforms. For the final stage, more advanced sensors and actuators are deployed for improving the successful outcome of the missions. In this situation, maximizing the data provided by the WSN becomes a critical factor and the WSN can offer support to the UAV platforms.

WSN and multi-UAV systems working together in disaster management applications are reviewed [123]. The associated wireless communication technologies are identified, from low-power Xbee links to various types of 802.11 Wi-Fi networks that provide increased bandwidth. The disaster management scenarios imply the possibility of dynamic reconfiguration of the roles of both the UAV and WSN networks depending on changing priorities in the field. These are focused on human-life protection through fast response while integrating the data sources that are available by means of intelligent aggregation and fusion mechanisms. WSN can be associated also with ground mobility platforms in the form of unmanned ground vehicles (UGV). Cloud infrastructures are able to support the large-scale data processing and optimization needed.

The AWARE platform proposed in [82], where WSN, BSN and UAV collaborate to gather data from harsh environments using both static and mobile sensors, gives a valuable insight on the capabilities and leveraged possibilities in this domain. Here UAVs are used as mobile nodes and for sensor deployment. The solution is capable to detect emergency events and to ensure autonomous network repair and a fast reaction.

### 6.4. Simulators for UAV–WSN Systems

The great majority of WSN–UAV interactions are done in simulation (comparatively few papers show real-life implementations). The simulations go from relatively simple (proof of concept validation done preponderantly in Matlab or similar scripting languages) to physics-based simulators, which aim to give an accurate representation of the WSN/UAV model in real-life functioning conditions. The last category uses popular tools like ROS (robot operating system) for modeling a control with simulation done, e.g., in Gazebo [124]. Other options are the flight simulator X-Plane [125], or, for the WSN interaction, the COOJA environment [126].

Note that, as far as we are aware, there is no tool that provides an integrated solution (at least not without a fair amount of tweaking) for both WSN and UAVs.

## 7. Discussion

The motion planning strategies employed in the papers studied for this review were mostly of relatively reduced complexity. Whereas the underlying optimization problem, which led to them, can be extremely complex, the paths are most often given as a sequence of segments linking consecutive waypoints (whose position and passing-through time are determined a priori).

While the complexity issues undoubtedly play a large part in the simplifications usually imposed (e.g., predefined types of paths), we believed that another reason for the current approaches is an attitude of “good enough” and “tested and true”. This is defensible in light of the currently available hardware resources (the UAV autopilot expects an ordered list of waypoints and the hardware itself is too expensive and/or dangerous to be tested on unproven strategies). Still, increases in the performance of inertial measuring units and the availability of global localization infrastructures (GPS, Galileo, etc.) mean that the practitioners can relax some of these constraints and propose more aggressive strategies (in the sense of reducing energy costs, operation time and the like).

Foremost, using realistic UAV dynamics in the motion planning procedure will give paths that are more flexible and can still be tracked by the autopilot at runtime with increased performance (tighter collision avoidance, less conservative energy estimation, reduced operation time, etc.). This is challenging from the viewpoint of the planning procedure (as the waypoint selection and subsequent ordering can no longer be separated easily but are realistic under the expected software and hardware improvements).

Another insufficiently treated aspect in the state of the art is the path planning in the multi-UAV context. Virtually all of the works studied first partition the tasks among the UAVs and then proceed to compute their paths independently. While this simplifies the analysis, it is conservative since it reduces the UAVs’ flexibility. Thus, computing overlapping paths or updating them online will allow a more flexible task allocation and will lead to an improved mission outcome.

Keeping in mind that future applications may work in an urban environment and that increased WSN interactions may require a low flight level, obstacle detection and subsequent avoidance as well as line-of-sight communication will have greater and greater importance in the path planning procedures. This will not only increase the complexity but will also lead to questions about safety regulations, reliable functioning (even in the presence of failures) and mission performance in a cluttered environment.

Overall, we consider that the field of motion planning, in the context of UAV–WSN applications, has many avenues of evolution and will evolve significantly in the coming years.

Figure 6 shows the overlapping areas of development in UAV, WSN and IoT systems by means of a Venn diagram. The key topics, also elaborated upon in the current articles, are highlighted. We consider the integrated heterogeneous monitoring systems that represent our focus to lay in the dashed area as illustrated in the figure. This represents the collaborative aspects of current UAV research topics such as: complex sensing, path planning and control over long distance links, in conjunction with typical WSN topics such as ground sensor deployment and coverage for data acquisition, energy efficiency and low power radio communication links. In addition, this core area is being extended to elements from the IoT domain by enabling the collaborative systems to leverage fog computing primitives, embedded data processing and inference mechanisms for local decision taking.

Regarding open research questions for integrated UAV–WSN systems, the current focus goes beyond optimal communication and data acquisition heuristics towards intelligent and adaptive systems. We predicted that this would be enabled by designing data and information flows across networks that allow distributed and localized inference at the network nodes. By adjusting such mechanisms to the specific resources and abilities of the nodes, the local decision is enabled. This improves the overall robustness and resilience of the large-scale monitoring system, allowing it to adjust to changing process dynamics and mission objectives, with minimal latency. An important requirement for this vision to be achieved is to make the intermediate gateways transparent at the upper levels of the system design in such a way that both low-level (ground) information as well as high level (inferred) information is available in both directions. Edge machine learning methods and techniques are an increasingly active field of study whose results can be successfully assessed in the context of integrated UAV–WSN systems.

Furthermore, on the communication aspects that enable these collaborative systems will need to be integrated into heterogeneous IoT solutions that offer seamless bidirectional data and information flows over multiple interfaces and network layers. New low power long range radio links such as LoRa or NB-IoT require different systems architecture for the design of large area monitoring systems, potentially enabling direct sensor to UAV integration. On the other hand, emerging technologies such as ultra-wide band (UWB), which enables precise local positioning in constrained environments can offer fine grained data to improve WSN–UAV interoperability.

One common challenge that has emerged from our study was the need for extensive scalability evaluations, which should prove the robustness of extended designed collaborative solutions. In practice that leads from current solutions with tens to hundreds of SNs and single or several UAVs to thousands of SNs and UAV swarms in seamless integration at the physical: sensing, communication, networking and logical levels: data, information and autonomous decision-making. This can be done through dedicated simulation environments and suitable test-beds.

## 8. Conclusions

The intelligent collaborative UAV–WSN systems have become the cheapest, friendliest and precise monitoring systems in various fields of economic or human interaction (environment surveillance, agriculture, smart cities, security, search and rescue missions, energy and transport infrastructure validation, etc.). Recent applications go beyond proof of concept systems and include multiple sensor nodes and UAVs, integrated into a large scale, geographically distributed, WSN. Due to the complexity and the multiple functionalities that need to be covered (acquisition, processing, collaboration and communication), these systems are, in essence, systems of systems. New concepts such as edge, fog and cloud computing are now used in data processing. The artificial intelligence of the basic elements (UAVs and SNs) makes the intersection between the entities in Figure 6 (UAV–WSN–IoT) increasingly relevant. Both UAVs and WSNs are increasingly considered as halves of a single multi-agent system, which functions organically as a single, indivisible unit. Not in the least, the nodes being part of a WSN are increasingly smart, with additional functionalities, which move them toward the IoT paradigm. Thus, we might conclude that systems integrating WSN, UAVs and IoT are on an ascendant trajectory and will become ubiquitous in commercial applications.

## Figures and Tables

**Figure 1 sensors-19-04690-f001:**
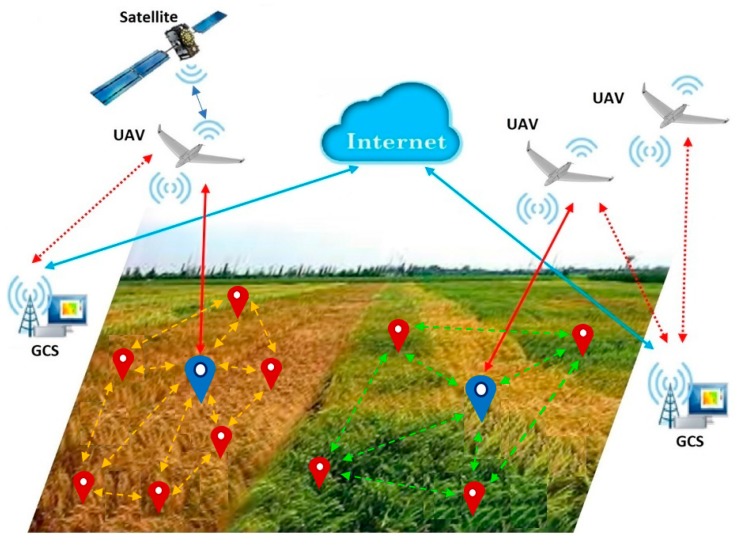
The concept of an integrated unmanned aerial vehicle-wireless sensor network (UAV–WSN) system.

**Figure 2 sensors-19-04690-f002:**
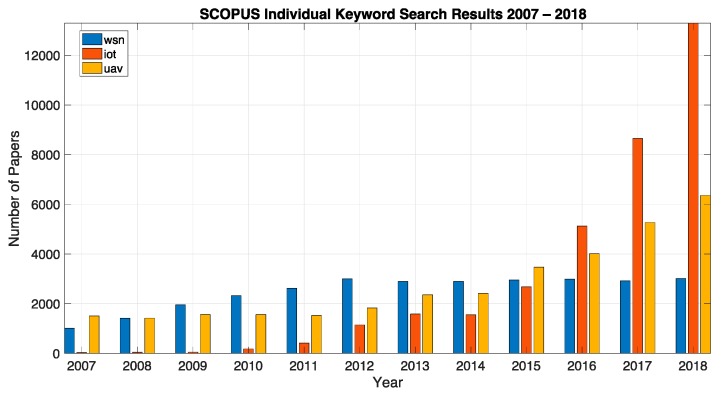
SCOPUS search results on keywords between 2007 and 2018: UAV, WSN and Internet of things (IoT), separately.

**Figure 3 sensors-19-04690-f003:**
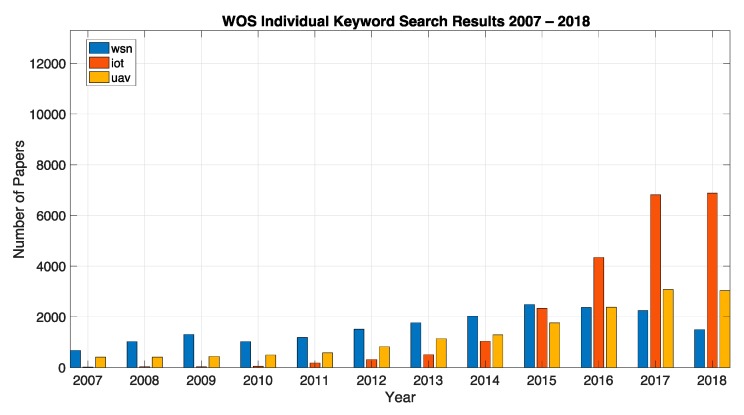
Web of Science (WOS) search results on keywords between 2007 and 2018: UAV, WSN and IoT, separately.

**Figure 4 sensors-19-04690-f004:**
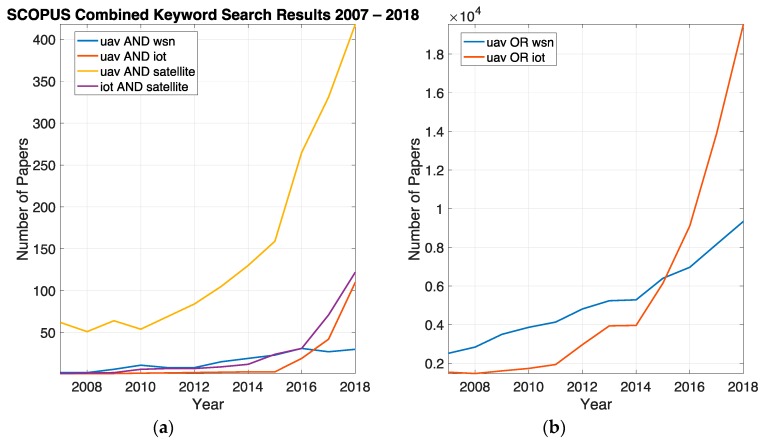
SCOPUS search results on keywords between 2007 and 2018: (**a**) UAV and WSN, together, and UAV and IoT, together and (**b**) UAV or WSN, UAV or IoT.

**Figure 5 sensors-19-04690-f005:**
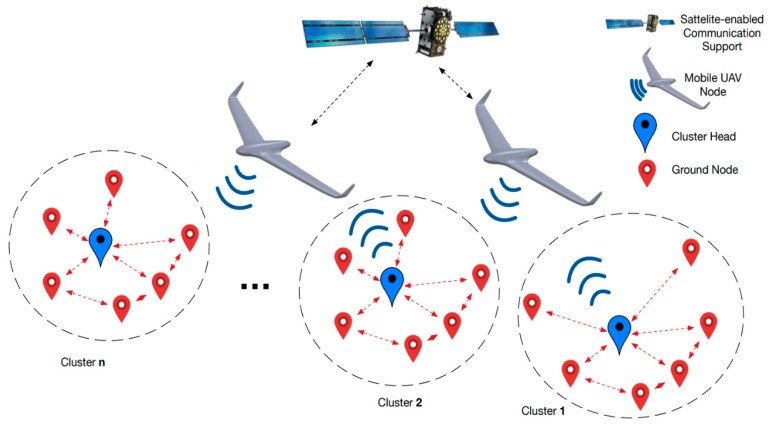
UAV data mule scenario among clusters of heterogeneous ground WSNs.

**Figure 6 sensors-19-04690-f006:**
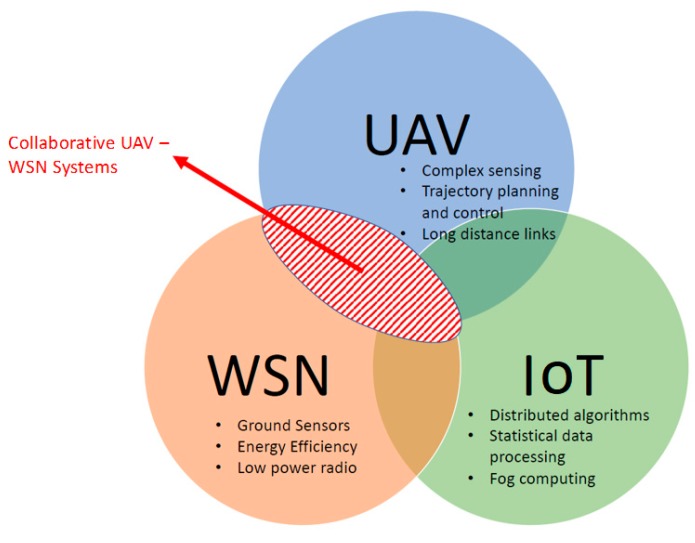
Research topics overlap in collaborative UAV and WSN systems with IoT aspects.

**Table 1 sensors-19-04690-t001:** Path planning for UAV–WSN systems.

Constraints and Costs	Method	Application	Miscellaneous	Reference
path length	GA	crop monitoring	cUAV, experimental results	[22]
WSN-based path update, communication radius, energy consumption	TSP	pesticide spraying, crop monitoring	cUAV, multi-pass, grid sensor deployment, experimental results	[19,24,25]
harsh-undulating terrain, communication radius, energy consumption	MINLP, TSP, FPPWR	data gathering	cUAV, forward and axial flight	[15,16]
path length, communication radius, energy consumption	greedy algorithm	data gathering	cUAV, experimental results	[50]
NA	Particle filtering, Bayesian analysis	data gathering	cUAV, node localization based on RSSI value, experimental testing	[45]
travel time, energy consumption	PSO, LEACH-C	data gathering	online CH selection	[55]
path length, energy consumption, communication range	heuristic methods, greedy algorithm	data gathering	spiral, zig-zag, strip-based paths	[17,18]
obstacle avoidance	MINLP, GA	area coverage, photogrammetry	mUAV, sensor placement	[28]
static/mobile obstacle avoidance	RRT, RRT*, GA	data gathering	mUAV, wUAV, experimental results	[46,56]
NA	heuristic algorithms, TSP	target tracking, area coverage	mUAV, experimental results	[48]
sensor lifetime	linear programming	data gathering	limited sensor buffer capacity	[21]
path length, travel time	heuristic methods	sensor node localization, data gathering	wUAV, multi-pass, experimental results, zig-zag path, sensor deployment	[47]
communication radius, path length, energy consumption	MINLP	data gathering	wUAV, multiple GDTs	[20]
obstacle avoidance, communication radius, path length, CH selection and estimation	MINLP, heuristic methods	precision agriculture	wUAV, B-spline parametrization	[1,57]
path length, area coverage	GA, PSO, Simulated Annealing, Hill-Climbing	pesticide spraying	experimental results	[49,58]
energy consumption, collision avoidance	MINLP, GA, RSCA, Set Cover Problem	data gathering	mUAV, TDMA, cyclic path	[59]
multiobjective utility function; prohibited, flying and sensing cells	GA, A* algorithm	data gathering over large areas	heterogeneous IoT devices	[60,61]
UAV payload and control restrictions	TTM	data gathering from sparse networks	experimental results, acoustic signals	[62]
path time, energy consumption	TSP, STTT	data gathering, recharging of depleted IoT devices	IoT devices	[44]

**Table 2 sensors-19-04690-t002:** Data acquisition and processing in integrated UAV–WSN systems.

Data Acquisition Mode	Data Processing Characteristics	Optimization/Intelligence	Implementation/Description	Reference
UAV as relay between disconnected WSN nodes	Local processing, Fog computing	Universal Plug and Play node discovery; services-based approach	Large scale IoT applications	[73]
WSN–UAV–Cloud	Fog computing	Area coverage	Distributed WSN	[50,74]
UAV as mobile sink	Local processing at UAV level	Increasing reliability, maximizing data collection	cUAV—multiple WSN, path model; UAV—ground Rician fading channel model; data collection model	[77]
WSN–UAV; UAV as mobile sink; WSN nodes with wake-up receivers	UAV local processing	A calibration step used for path planning, a probability map for data collection is built	Small scale WSN	[78,79]
WSN–UAV; WSN with both normal and GPS sensors; UAV as mobile sink	Node localization, grid division and path planning at base station UAV local processing	Aerial node deployment Node localization Flying path optimization	Large scale WSN	[16,18]
WSN–multiple wUAV UAV as mobile sink	UAV local processing	Path planning, CDR (Conflict Detection and Resolution)	Multiple small scale WSN; Matlab simulations for trajectory; Real experiments using Megastar wUAV	[46]
UAV as relay between cluster head and GCS	Only messaging for data acquisition is considered at WSN and UAV	Energy consumption optimization	Mathematic simulations evaluating the effect of distance between cluster head and base station	[38]
UAV as relay between WSN cluster heads	Only messaging for data communication is considered at WSN and UAV	Method for choosing the cluster head and routing protocol	Sparse WSNs with unbalanced traffic	[64]
UAV as relay between WSN cluster heads	Only messaging for data communication is considered at WSN and UAV	Maximize area coverage	Sparse WSNs	[26]
Multiple UAVs as relays between WSN cluster heads	Path computing at UAV level	Different messaging architectures Minimization of the sum of all distances	Sparse WSNs; Mathematical simulations evaluating different architectures and path planning methods	[39]
UAV as relay between WSN cluster heads and sink nodes	Only messaging for data communication is considered at WSN and UAV	Reduce the consumption in data transmission; framework for monitoring linear infrastructure	Linear (deployed) sensor network with sink at the end line.	[40]
Multiple UAVs as relays between WSN nodes	Only messaging for data communication is considered at WSN and UAV	UAV positioning	Multiple faults in linear WSN	[42]
Multiple UAVs as mobile nodes	UAV route processing	Randomly selected routes Full area coverage	Mathematical analysis of the area coverage with the proposed algorithm	[41]
Multiple UAV- WSN; UAV as mobile sink	Data aggregation	Path planning UAV energy optimization	Large area WSN with scattered nodes; OmNET++Simulator	[59]
WSN–UAV–IoT	Data aggregation; Animal movement prediction; UAV route processing	Path planning defined by external factors	Large area WSN; Simulations using Zebranet dataset	[80]
	Data aggregation UAV route processing	Path design to four functions: sensing, energy, time, and risk	Large area WSN	[60]
	Local processing, cloud computing	Energy consumption	Farm beats gateway for data collection from WSN, UAV; Azure Cloud	[81]
UAV for WSN node localization	UAV route processing	Node position estimation Path planning	Large scale WSN; Mica2 Crossbow as WSN nodes	[45]
UAV for WSN node deployment	Messaging for status communication is considered at WSN and UAV	Node deployment reliability	AVATAR Autonomous helicopter; Mica Motes for WSN nodes	[30]
WSN–UAV–BSN	Messaging for status and data communication	Latency, reliability, network dynamics	OmNET++Simulator	[82]

**Table 3 sensors-19-04690-t003:** Data communication in integrated UAV–WSN systems.

Data Communication Type	Standards (Implementation)	Details	Reference
WSN–mUAV–fog	GSM, UMTS, HSPA	Standards are evaluated considering a bandwidth for sending video data from 1000 Kbps to 6000 Kbps	[73]
wUAV–WSN (CHs)	ZigBee	Distance between UAV and WSN below 100 m for communication	[95]
UAV–WSN UAV as mobile sink	ZigBee	Nodes in stand-by mode; WSN nodes with wake-up receivers; Point-to-point wake-up	[79]
UAV–IoT	IEEE 802.11 (WiFi) For wide areas: LoRaWAN, LTE, LTE-A, IEEE 802.16 (WiMAX)	Data collection and energy harvesting	[44]
UAV–WSN	LoRa	Distance between UAV and WSN 4 km	[10]
UAV–WSN	TinyMesh	Distance between UAV and WSN 485 m	[96]
UAV–WSN	RF	Distance between UAV and WSN 1–2 km	[97]
UAV–WSN	RF@902-928 MHz	200 m in NLoS conditions	[62]
UAV–WSN	RF@900 MHz	UAV broadcasts wake-up messages every 1 s. UAV speed and ground distance not available	[45]
UAV–WSN	RF@916 MHz	Maximum air-ground distance 13 m, median range 9 m	[30]
UAV–WSN (single hop)	BLE	Nodes wake up each 10 s Distance between 10 m and 20 m	[98]
cUAV–mWSN	CSMA/ CD/ IEEE 802.15.4	Nodes in stand-by mode, wake up for sensing or for UAV data transmission, distance between 10 m and 30 m	[22]
cUAV–WSN	IEEE 802.15.4g for data communications (920 MHz) IEEE 802.11n@5GHz for UAV ground control	Different wake-up mechanisms: broadcast and unicast, nodes in stand-by mode, a wake-up receiver installed at WSN nodes	[78]
UAV–WSN	IEEE 802.15.4 for data acquisition from WSN, 6LoWPAN for data sinking	Dual stack single radio architecture, algorithm for improving data transmission reliability	[71]
UAV–UAV	UHF band (400 MHz)	Direct communication, in near line of sight or in non-line of sight conditions	[90]
cUAV–cUAV	OLSR dynamic protocol, Ad-hoc Wi-Fi infrastructure	One UAV serves as a relay between another UAV and GCS	[92]
cUAV–cUAV	RTSP and RTP protocols	Video processing pipeline and control, quad copter AR Drone	[93]
cUAV–GCS–cUAV	4G/LTE, DR 915 MHz Radio Telemetry (UAV-GCS), IEEE 802.3 32-bit CRC polynomial	Indirect communication, connectivity for low altitude UAV through a terrestrial 4G/LTE network	[91]

**Table 4 sensors-19-04690-t004:** Integrated UAV–WSN systems.

System Architecture	System Configuration	System Implementation	Reference
UAV–WSN	cUAV, wUAV—single WSN	Quadrotor	[22]
	Miniature UAV—single WSN	Hero 6 UAV, Mica2 WSN nodes	[45]
	UAV for WSN node deployment	AVATAR helicopter, Mica Mote WSN nodes	[30]
	cUAV–WSN	Different simulators: OmNET++, Glomosim, SSFNet, ns2, Java-Sim	[24]
	wUAV–WSN	Fury UAV	[62]
	mUAV with specific functions: mobile sink, node deployment	OmNET++ simulations	[82]
	Multiple wUAV–WSN	Simulations using Matlab and C++, real evaluation using Megastar wUAV	[46,56]
	wUAV–WSN	Matlab simulation	[100]
	wUAV–WSN (single hop)	DUNE software to communicate with X8 UAV Pandaboard at UAV level ATXMEGA192C3 for sensor node	[97]
	wUAV–WSN (single hop)	PIC24F PCB as sensor node BeagleBone Black (BBB) board as beacon node, buoy, Delta wUAV	[10]
	wUAV–WSN (single hop)	Silicon Labs EFM32 Gecko microcontroller as WSN node, TBR-700 receiver and X8 UAV for field tests, Phantom quadrocopter for off-shore tests	[96]
	cUAV–WSN	DJI Phantom 4 UAV, embedded controller for WSN nodes, PC with an Android terminal for ground control	[78]
UAV–WSN–IoT	mUAV–WSN	OmNET++Simulator	[108]
	cUAV	Simulations using Arduino boards for both WSN and UAV	[73]
	cUAV–WSN	Arduino, Particle Photon or NodeMCU boards at sensor level, embedded Farmbeats gateway at UAV level, DJI Phantom 2, Phantom 3 and Inspire UAVs, Raspberry Pi at base station, Azure Cloud	[81]

**Table 5 sensors-19-04690-t005:** Main applications of the integrated UAV–WSN systems.

Application Domain	Application Details	Implementation Characteristics	Reference
Agriculture	Data driven, IoT base station, prediction models	Farm beats gateway for data collection from WSN, UAV Azure Cloud	[81]
	Crop monitoring in vineyards	Quadrotor	[22]
	Pesticide spraying	Simulations with OMNeT++ and MiXiM	[24,37,58]
	Precision agriculture	cUAV, multi-pass, grid sensor deployment, experimental results	[25]
	Crop and soil monitoring in vineyards integrating environmental data with multispectral images	mUAV WSN node: Arduino	[23,109,110]
	Farmland environmental monitoring	WSN sensor and relay nodes Octocopter	[111]
Environment monitoring	Underwater monitoring	Simulations using Arduino boards	[73]
	Ambient air pollution monitoring	Quadrocopter	[112]
	Ambient monitoring: temperature, humidity, light intensity, wind speed	Fixed-wing UAV	[90]
	Marine environment monitoring	PIC24F PCB/ ATXMEGA192C3 as sensor node, BeagleBone Black (BBB) board as beacon node, buoy, Delta mUAV	[10,96,97]
Animal monitoring	Endangered species movement monitoring without any attached devices	Simulations using Zebranet dataset	[80,113]
Disaster monitoring/ emergency	Situational awareness, pre-event and post-event activity functions, integration with BSN	OmNET++ Simulator	[82]
	Disaster monitoring, situational awareness, pre-event and post-event activity functions	Framework model	[114]
	Natural disaster management	Evaluation of different implementation solutions	[115]
	Disaster recovery, post-event functions	Quadrocopter	[116]
	Situational awareness, post-event activity functions	Numerical simulations of communication cases	[117]
	Disaster monitoring, situational awareness	Framework architecture	[118]
	Flood monitoring, post-event functions	Quadrocopter	[119]
Transport monitoring	Ground pipeline monitoring and control	Simulations using Arduino boards	[73]
	Energy-efficient urban surveillance, Intelligent transportation system	Multicopters in LoRaWAN-like networks	[120]
Utilities	Power meter reading	Sinalgo Simulator, scenarios with up to 16,000 nodes	[53]
Multimedia	Capturing HD/3D content with augmented reality	Platform description	[121]

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
