# Peer review of "A Survey of Collaborative UAV–WSN Systems for Efficient Monitoring"

_sensors, 2019, doi:10.3390/s19214690_

Round 1
Reviewer 1 Report
On the positive side, the authors have spent an extensive amount of work looking through a wide and diverse literature, there might be a value of such an overview paper. However, I feel that the paper is generally poorly presented and lack of focus, and because it covers so many different topics, it doesn't give enough depth about each topic, and as such it does not have enough value compared to more focus survey paper. There would be value for such a wide overview paper if it was focused on how the different component integrate and interact with each other.
Discussion
This is essentially an heterogenous multi-agent system, I fiind it odd that the word agent is mentioned only four times, and "multi-agent" only once on line 721. I would have expect to find that in the introduction and through the paper. The heterogenous aspect is only mentioned in the discussion section.
While I appreciate the idea of environmentally friendly UAVs, I am not entirely sure how that is relevant and how "gaz" powered UAVs would behave differently in such a system. On a side note, satellites are not entirely environmentally friendly, and their impact on the environment is getting more and more controversial (especially due to the release of small particles [1]).
I can agree that IOT branched out of sensor networks, but it actually branched out when we realized that the devices could be used for more than sensing, and as such, I am not sure it is a valid keyword to use for comparison of search in section 2.
Suggestion of improvement
Improvement for the content
l32:I would like some examples of the advantages listed, generally speaking, when citing a paper, even in the introduction, I expect to find some information about the content of the paper and not just the number. Same apply to l53-54. l75: the transition to path planning was very abrupt. After reading the paper, path planning is a smaller part of the survey, why is it more prominently mentioned? There is no general overview of the paper l104: if you neglect the entire field of multi-agent systems... l161: UAVs can also be used for deployment of WSN
Improve the language
Generally speaking, I think the paper is poorly written, and I would suggest the author to review the text more thoroughly. Bellow is a limited set of suggestion of improvement:
l16 "a comparative perspective"? l48-51 is difficult to read and understand l33 "the WSN" -> WSN l156-157, unclear what is the feedback l721 multi-agent And so on...
References
[1] Potential climate impact of black carbon emitted by rockets, Martin Ross,1Michael Mills,2and Darin Toohey, GEOPHYSICAL RESEARCH LETTERS, VOL. 37, 2010
Author Response
We thank the reviewer for its comments and constructive observations, which have led to a hopeful, improved paper. We believe that the new changes and comments included in the manuscript should fully satisfy the points raised by the reviewer. We tried to improve the content to better explain the meaning of the article.
Please see the attachment.

Reviewer 2 Report
This paper presents an interesting scope, which is providing a systematic literature review for UAV-WSN systems for (Efficient) monitoring. It is a quite challenging goal, since as authors noticed there are lots of scientific work in this area.
My first concern with this paper is regarding lack of detailing for the adopted “systematic” review protocol. For instance, what is the adopted criteria for selecting or rejecting a paper for analysis? How do authors organize the selection? Besides, I think that only working with three keywords (UAV, WSN, and IoT) is too restrictive. I see interesting papers not listed in the analysis, for instance:
Sayyed, Ali; DE ARAÚJO, GUSTAVO; BODANESE, JOÃO; BECKER, LEANDRO; Dual-Stack Single-Radio Communication Architecture for UAV Acting As a Mobile Node to Collect Data in WSNs, Sensors (Basel). v. 15. p. 23376-23401, 2015
Also, the content regarding each selected paper could be improved in my perspective. Also, the content from sect. 4 (Discussion) and 5 (Conclusion) is vague and must be improved.
I do not think that Table 1 is adequate for a journal publication. Also, figs 1 and 6 can be improved.
Author Response

(The authors gave the same response as above.)

Reviewer 3 Report
Paper shows a comprehensive literature review of WSN with UAVs as data mules. Problems surveyed are mainly UAV route planning, communication between WSN and UAV and data aggregation.
Some minors/typos:
L125 UAV and WSN -> UAV and IOT
L148 Caption should be forced near the picture.
L151 Not sure that research is the right word for this. Maybe study or assessed?
L334 mehtods
L353 and -> an
L521 Table header "Reference" hifen
Some doubts:
L360 Disagree. The WSN still have to keep those protocols to reach de CH.
L379 What do you mean by "infrastructure less integration"?
Overall, paper is well written and covers most part of the advance in the fields on the last 10 years. I would like to suggest a section expansion including only data-mule protocols, since they can be direct mapped onto your scenario. Please also dedicate a small section tabulating the simulators, since there are a myriad of them and most part of the works are simulated. It would be nice to explain how researchers are simulating UAV-WSN, giving that there is no simulator to this specific scenario afaik.
Author Response

(The authors gave the same response as above.)

Round 2
Reviewer 2 Report
I really appreciated authors efforts to improve the paper. All recommendations were properly addressed, and I am pleased with that. I consider that the paper quality considerably improved.
Author Response
Dear reviewer,
Thank you very much for the relevant comments that helped us improve the article.
All the best,
Dan Popescu